# NH_3_ Sensor Based on rGO-PANI Composite with Improved Sensitivity

**DOI:** 10.3390/s21154947

**Published:** 2021-07-21

**Authors:** Fabio Seiti Hadano, Anderson Emanuel Ximim Gavim, Josiani Cristina Stefanelo, Sara Luiza Gusso, Andreia Gerniski Macedo, Paula Cristina Rodrigues, Abd. Rashid bin Mohd Yusoff, Fabio Kurt Schneider, Jeferson Ferreira de Deus, Wilson José da Silva

**Affiliations:** 1Graduate Program in Electrical and Computer Engineering, Federal University of Technology—Paraná, Curitiba 80230-901, Brazil; fabiohadano@alunos.utfpr.edu.br (F.S.H.); gavim@alunos.utfpr.edu.br (A.E.X.G.); fabioks@utfpr.edu.br (F.K.S.); 2Institute of Physics, University of São Paulo, São Carlos 13566-590, Brazil; josiani@ifsc.usp.br; 3Graduate Program in Physics and Astronomy, Federal University of Technology—Paraná, Curitiba 80230-901, Brazil; saragusso@alunos.utfpr.edu.br (S.L.G.); agmacedo@utfpr.edu.br (A.G.M.); jfdeus@utfpr.edu.br (J.F.d.D.); 4Graduate Program in Chemistry, Federal University of Technology—Paraná, Curitiba 81280-340, Brazil; paulac@utfpr.edu.br; 5Department of Physics, Vivian Tower, Singleton Park, Swansea SA2 8PP, UK; a.r.mohdyusoff@swansea.ac.uk

**Keywords:** rGO, PANI, ammonia, clusters, NMP, carboxylic, acids

## Abstract

This work reports on a reduced graphene oxide and poly(aniline) composite (rGO-PANI), with rGO clusters inserted between PANI chains. These clusters were formed due the plasticizing effect of N-methyl-2-pyrrolidone (NMP) solvent, which was added during the synthesis. Further, this composite was processed as thin film onto an interdigitated electrode array and used as the sensitive layer for ammonia gas, presenting sensitivity of 250% at 100 ppm, a response time of 97 s, and a lowest detection limit of 5 ppm. The PANI deprotonation process, upon exposure to NH_3_, rGO, also contributed by improving the sensitivity due its higher surface area and the presence of carboxylic acids. This allowed for the interaction between the hydrogen of NH_3_ (nucleophilic character) and the -COOH groups (electrophilic character) from the rGO surface, thereby introducing a promising sensing composite for amine-based gases.

## 1. Introduction

Recent advances have been achieved in the field of nanostructured devices for energy conversion and harvesting [1,2,3], motion [4], or physiological monitoring [5] due to enhanced or new properties observed in materials processed at the nano-scale. Poly (aniline) (PANI) was used as a sensitive layer for detection of ammonia gas due its reversible transduction mechanism, combined with high sensibility and selectivity when exposed to NH_3_ molecules [6,7,8,9]. These devices have been used to monitor NH_3_ in several environments, such as in ambient atmosphere (0.1 ppb until >200 ppm) [10,11], animals stalls (1 ppm until >25 ppm) [12,13], vehicles (4 g/mi–2000 g/mi) [6,14], leakages (200 ppm–1000 ppm) [15], and the breath process (50 ppb–2000 ppb) [8,16]. Therefore, development of portable NH_3_ gas sensors produced with new composites, which present proper response and stability in ambient conditions, are still interesting for several applications.

The acid-base reaction between the protonated (emeraldine salt—PANI-ES) and deprotonated (emeraldine base—PANI-EB) forms of PANI is a simple method used for detecting NH_3_ molecules. When exposed to ammonia vapors, the PANI-ES undergoes deprotonation, generating PANI-EB, which has lower electrical conductivity [17,18]. Moreover, in order to improve the sensing capabilities, several chemical modifications or different morphologies were reported, such as sulfonation of the benzene rings and production of nanowires [19], nanofibers [20], or nanoparticles [21].

Reduced graphene oxide (rGO) presents bidimensional carbon atoms linked through sp^2^ bonds with high electrical conductivity, up to 6000 S/cm, and proper mechanical features [9,22,23]. Therefore, rGO has been used as an electrode in organic based devices [23,24], chemical sensors [25], and field-effect transistors (FET) [26], among others. For gas sensing, rGO has a high surface area available to interact with the environment, thus increasing the device sensing performance for several gases [27,28,29]. Enhanced electronic properties of graphene derivatives have been obtained through methods such as partial reduction, and thermal annealing or surface functionalization by non-covalent/covalent bonding [30,31,32,33,34]. The latter method requires covalent chemical bonds at the basal plane of carbon layers, which results in an alteration of electronic features, increased solubility, and processability as thin films [30]. This procedure has been used to produce rGO composites with fluorine/chlorine atoms [31,32], ammonia [32], and PANI [34].

PANI interacts with carbon-based nanomaterials containing sp^2^ bonds by π-π interactions [35,36], allowing for processes of charge transfer and, thus, increasing the delocalization of the charge carriers. Therefore, rGO-PANI composites have resulted in an improved performance as an electrical sensing layer [6,22]. For instance, Gang et al. reported stretchable skins produced with rGO and PANI nanowires for the fabrication of pressure sensors [37]. Wu et al. showed that the lowest NH_3_ detection limit of a GO-PANI composite was lower than in PANI-based sensors, being 1 ppm and 10 ppm, respectively, but with low sensitivity around 3.5% [18]. Bai et al. reported on an rGO-PANI hybrid synthesized through chemical oxidative polymerization, with the composite deposited onto a PET/ITO substrate and tested for NH_3_ sensing at room temperature, the results of which showed a sensitivity of 350% at 100 ppm of NH_3_ [38]. Huang et al. reported on the use of rGO-MnO_2_ as a template and oxidant for an aniline monomer, which resulted in PANI nanoparticles anchored onto the surface of rGO sheets, a composite which presented a sensitivity of 45% for NH_3_ gas [22]. The processes regarding the interaction between NH_3_ molecules and the rGO-PANI composite are presented in Figure 1, where NH_3_ molecules interact with the rGO-PANI composite, inducing a phase conversion of PANI from the ES to the BE phase [13]. This deprotonation process resulted in PANI and rGO-PANI with a lower conductivity. Moreover, rGO also contributes to the sensing process since its electrical resistance increases upon NH_3_ exposition, mainly due to an interaction between the H of NH_3_ (nucleophilic character) and the -COOH groups (electrophilic character) [39] present on the surface of rGO [40] and the increased surface area [27,28,29].

Herein, rGO powder was prepared through a modified Hummers method. Spherical clusters of rGO were inserted between the PANI chains; these clusters were formed due to the plasticizing effect of N-methyl-2-pyrrolidone (NMP) solvent, which was added during the preparation of this composite. This rGO-PANI composite was tested as an NH_3_ sensor in a closed environment with a controlled amount of NH_3_, and presented with proper sensitivity when compared with similar sensors reported in the literature.

## 2. Materials and Methods

PANI-ES (M_w_ > g mol^−1^), ammonium hydroxide solution (NH_4_OH, 28% NH_3_), and polyethylene terephthalate (PET) substrate coated with indium tin oxide (ITO, 60 Ω/sq) were purchased from Sigma Aldrich^®^ (San Luis, EUA). All reagents were ACS grade and were used without purification.

Graphene oxide (GO) was obtained through a modified Hummers method [41] as follows: graphite powder was mixed with sodium nitrate (NaSO_4_) on a 2:1 weight ratio and stirred in a sulfuric acid (H_2_SO_4_) solution in an ice bath. During the agitation process, potassium permanganate (KMmO_4_) was slowly added to the dispersion. Then, the ice bath was removed, and the temperature was kept at 35 °C for 30 min. Further, deionized water was added and, after 15 min, the suspension was filtered and washed four times with water and dried at 60 °C in a vacuum oven. The reduction of the GO sheets was performed by heating the resulting powder up to 450 °C over 2 h. The PANI-ES powder was dissolved in NMP with a concentration of 20 mg mL^−1^ upon magnetic stirring over a period of 15 min, then, 15 µL of NH_4_OH was added into the PANI solution remaining. Afterwards, rGO was added into the PANI solution at a 1:5 weight ratio and remained upon 3 h of ultrasonic stirring.

The sensors were produced by spin coating 20 µL of rGO-PANI dispersion onto an interdigitated electrode array at 1500 rpm for 1 min. Subsequently, the devices were treated by thermal annealing at 80 °C for 2 h.

Scanning electronic microscopy (SEM) images were obtained with an EVO^®^ MA15 (Curitiba, Brazil) at 20 kV, and atomic force microscopy (AFM) images were obtained using a Shimadzu^®^ SPM 9500 J3 (Curitiba, Brazil) using the non-contact mode. FTIR and UV-visible spectra were acquired in a Varian model 640 spectrometer using an attenuated total reflection (ATR) mode and a Kasuaki^®^ IL-592 spectrophotometer (Curitiba, Brazil), respectively. Impedance spectroscopy was performed in a Solartron^®^ 1260 (São Carlos, Brazil) connected with a dielectric interface 1296, a Nyquist plot was performed using the EIS Spectrum Analyzer software [42], and additional electrical measurements were performed in a Keithley^®^ 4200-SCS (Curitiba, Brazil) semiconductor parameter analyzer.

In order to evaluate the sensor response, the device was isolated in a closed environment chamber containing NH_3_ gas and connected to the Keithley^®^ 4200-SCS equipment through a four-probe setup. A constant DC voltage (0.15 V) was applied to monitor the resistance against the time of exposure to NH_3_. The variations of resistance in relation to the reference was calculated using Equation (1):(1)$$\Delta R\left(\%\right)=\left(\frac{R_{air}-R_{gas}}{R_{air}}\right)\times 100$$
where Δ*R*, *R_air_*, and *R_gas_* are the percent variation of resistance, resistance in air, or NH_3_ gas, respectively. The analyte was inserted into the chamber by using chromatography syringes with adjustable volumes of 0.5, 1, and 5 µL. The NH_3_ concentration (in ppm) was calculated using Equation (2):(2)$${\left[NH_{3}\right]}_{ppm}=\frac{\rho VRT}{MpV_{c}}\times 10^{6}$$
where ρ, *V*, and *M* are density, volume, and molar mass of the analyte, *R* is the ideal gas constant, *T* is temperature, *p* is pressure, and *V_c_* is chamber volume.

## 3. Results

### 3.1. Morphology

Thin films of PANI (in the emeraldine salt phase) were produced for comparing properties and sensor response with those acquired from rGO-PANI composite. Figure 2 shows the corresponding SEM images acquired from the PANI and rGO-PANI films. The image acquired from the PANI film demonstrated the presence of particles (chain aggregates) with an average diameter of 450 ± 100 nm. Normally, PANI films prepared from NMP solutions contain a considerable amount of solvent, about 18% by weight, arising from the high boiling point of the solvent (202 °C) and the presence of the hydrogen bonding of the carbonyl group with the amine group in PANI [43]. Moreover, Zheng et al. showed that this interaction resulted in particles with amorphous character and reduced electrical conductivity and, thus, the insertion of rGO into the PANI solution can enhance the electrical features and result in interconnected PANI particles [44,45]. In fact, these features are observed when Figure 2a,b are compared, as in the rGO-PANI composite, the PANI particles were more interconnected than in the PANI polymer. Moreover, flower-like structures of rGO sheets, with diameters of ca. 2 µm, were identified in Figure 2b, surrounding the PANI particles. Furthermore, recent literature highlighted a gain in the surface area in composites containing rGO or GO due their layered structures [46,47].

AFM images are displayed in Figure 3. The PANI film had a root mean square roughness (R_rms_) of 75 nm, resulting from the chain aggregates [44,48]. The surface of rGO-PANI film presented R_rms_ of 20 nm, while the corresponding force mode image showed regions with different contrasts, arising from regions with different mechanical features related to the PANI and rGO-rich PANI surfaces, illustrating the distribution of rGO along the PANI surface.

### 3.2. Spectroscopy Characterization

Figure 4 presents the UV-vis absorbance spectra acquired from PANI and rGO-PANI.

The absorbance spectrum acquired from rGO-PANI in the NMP suspension presented two broad bands with maxima at 350 nm and 675 nm, attributed to π − π* transition and from the exciton transition of the quinoid rings, respectively [49,50]. This spectrum was red-shifted when compared with the absorbance acquired from PANI. This effect arose from the interaction between the PANI and rGO sheets in the composite, which increased the delocalization of electrons along the polymer backbone chain and resulted in increased electrical conductivity [51]. At the solid state (thin film), there was an additional peak localized at ca. 440 nm, which could be assigned to the polaron–π* transition of PANI, suggesting that the imine nitrogen was protonated by the carboxylic acids present and effectively anchored onto the surface of rGO sheets due to the conversion of these segments into poly(semiquinone) cation-radicals [22,52,53,54]. The values of energy band gaps were estimated using the Tauc plot method, resulting in 3.2 eV and 3.1 eV for PANI and rGO-PANI, respectively, which could be attributed to the presence of defects from rGO and higher conductivity [55,56].

The FTIR spectrum of the rGO-PANI film is exhibited in Figure 5. The bands at 3306, 1662, 1595, 1494, 1389, and 1250 cm^−1^ correspond to -OH and -NH, C=O (from rGO and NMP), C=C (quinoid rings), C=C (benzenoid rings), -OH in C-OH, and CN, respectively. Due to the interactions (π − π interaction and hydrogen bonding) between PANI and rGO, the bands related to the PANI aromatic rings and quinoids, as well as the NH band, suffered a shift towards higher wavenumbers [57]. The presence of residual NMP in the film was confirmed through the band in 1662 cm^−1^ [43].

### 3.3. Electrical Characterization

Electrical features of the rGO-PANI composite were evaluated through a four-probe method and impedance spectroscopy. Pristine PANI-EB presented an isolating character with sheet resistance of ~260 MΩ/sq, but this parameter was reduced to 43 MΩ/sq at the rGO-PANI (1:5, wt:wt) due the protonation of the imine nitrogen at PANI-EB and increased conducting paths in rGO-rich regions. Moreover, PANI can interact with carbon-based materials containing sp2 bonds via π−π interactions between the aromatic and quinoid rings of PANI and the π-electrons in graphene [6]. This feature mediates charge transfer processes and increased the charge delocalization at the PANI chains and, thus, resulted in higher electrical conductivity [35]. Further, in order to investigate these electrical features, PANI and rGO-PANI films were analyzed by EIS measurements at a frequency range from 1 Hz up to 10 MHz (amplitude of 10 mV). The results and the Nyquist plot [2] are presented in Figure 6.

The Nyquist plot presents two semicircles arising from a series resistance, attributed to interfaces with the electrodes (denoted as R_1_), and a shunt resistance, related to the interfaces along the film (denoted as R_2_). Theoretical fitting from the equivalent circuit to the experimental data showed that the R_1_ parameter was negligible in both materials (PANI and rGO-PANI) due low values of contact resistance with the electrodes. Therefore, the impedance of the device was mainly determined by the R_2_ parameter, which was quite dependent on the conducting regions present in the film. In rGO-PANI, the presence of conducting rGO clusters between the PANI resulted in a lower R_2_ parameter, with R_2_ values of 22.10 MΩ and 9.40 MΩ in PANI and rGO-PANI, respectively. Similar results were observed in rGO-PANI composites [58] and nanohybrids [34]. Moreover, the equivalent circuit presented a constant phase element (CPE), denoted as Q_2_, which represented a non-ideal capacitor arising from the non-uniform distribution along the film [59].

### 3.4. Evaluation of Sensor Response to NH_3_

The sensor response for NH_3_ molecules was evaluated through monitoring variations in electrical resistance upon exposure and after NH_3_ removal. Figure 7a presents the sensitivity, obtained through Equation (1), against time. These curves showed that the rGO-PANI-based sensor showed fast response when compared with the PANI-based sensor. The response and recovery times were 97 s and 680 s, respectively, being at the same order as values reported in literature [60,61]. These values were shorter when compared with the PANI-based sensor, which showed response and recovery times of 190 s and 766 s, respectively. The enhanced sensing response observed in the rGO-PANI composite may arise from the synergistic effect of the protonation process in rGO-PANI, as both PANI and rGO materials have functional groups which may interact with NH_3_ molecules. In addition to the NH_3_ protonation process by PANI-ES (Figure 1), carboxylic acid groups from the rGO surface also contributed to the formation of the NH^4+^ ion, as suggested by Figure 1. This synergistic effect resulted in the fast variation of the electrical resistance upon NH_3_ exposure, and both processes would be reversible, in principle. However, the PANI-based sensor presented a partial reversibility which may be related to the retention of ammonium cations in the film surface, while the rGO-PANI-based devices presented a complete reversibility after multiple cycles in 600 ppm of NH_3_.

Moreover, calibration curves for sensitivity against the concentration of NH_3_ were acquired from both devices (Figure 7b). At the range between 5 ppm and 100 ppm, both PANI and rGO-PANI sensors showed a linear response with the NH_3_ concentration, with slopes of 1.5 and ~3.0, respectively. In this case, the increased slope indicated sensors with improved response, which can provide more accurate results.

These devices were monitored over 2 months in air. After 4 days, the sensor presented a decrease in the sensitivity from ~350% to 140% (at 600 ppm) and, after this reduction, remained constant over time. This result may have arisen from the aging of the composite. Table 1 summarizes the main parameters reported from NH_3_ sensors in the literature. The rGO-PANI-based sensors presented superior sensitivity when compared with composites containing GO or nanoparticles, which may arise from the functional groups present at rGO surface that potentialize the interaction with NH_3_. Moreover, this composite could detect the presence of NH_3_ from a concentration of 5 ppm, and this value and the high sensitivity represent a gain when compared with the response of PANI-based sensors.

## 4. Discussion

Therefore, the rGO-PANI reported in this work is a promising NH_3_ sensing material. It was prepared through a two steps synthesis: (1) a modified Hummers method was used to produce rGO, and (2) rGO was mixed with PANI in an NMP suspension. Despite higher values of response and recovery times, this rGO-PANI-based sensor detected lower amounts of NH_3_ (5 ppm) when compared with the sensor containing the rGO-PANI composite prepared in one step synthesis (10 ppm) [38]. Therefore, new routes for synthesis or surface functionalization may allow for the production of sensors that detect low concentrations and increase the detection range of NH_3_. As a future work, changes in the rGO:PANI ratio, which in this work was fixed at 1:5, could be performed in order to increase the sensitivity and reduce the detection limit for NH_3_.

## 5. Conclusions

In summary, an rGO-PANI composite was prepared in a two steps routine, resulting in rGO clusters dispersed between interconnected PANI particles, with reduced electrical resistance when compared with PANI. The values of energy band gaps were 3.2 eV and 3.1 eV for PANI and rGO-PANI, respectively, and this slight increase was attributed to the presence of defects from rGO and the higher conductivity in the composite. EIS characterization also showed lower shunt resistance in the rGO-PANI films, as well as low contact resistance with the electrodes. Applied as the NH_3_ sensing layer, the rGO-PANI composite presented fast response and suitable reversibility when compared with the PANI-based device, with a sensitivity of 250% at 100 ppm, a response time of 97 s, and a lowest detection limit of 5 ppm. These results illustrate that new routes for synthesis or surface functionalization can result in sensors with increased detection range and high sensitivity, as required in commercial applications.

## Figures and Tables

**Figure 1 sensors-21-04947-f001:**
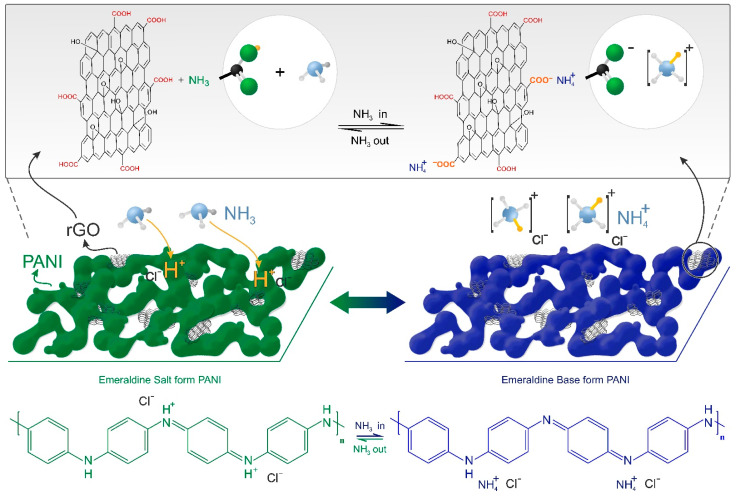
NH_3_ protonation/deprotonation process in PANI chains and rGO sheets.

**Figure 2 sensors-21-04947-f002:**
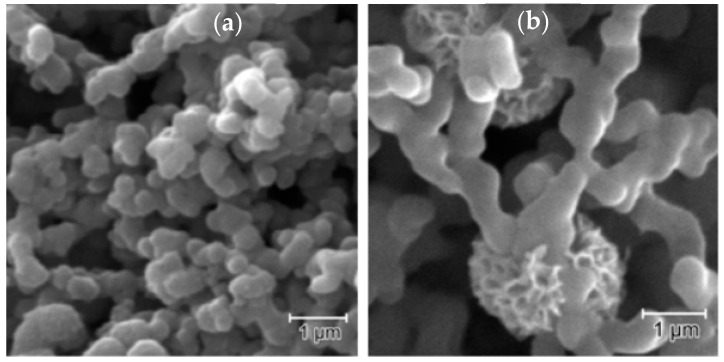
SEM images acquired from (**a**) PANI and (**b**) rGO-PANI.

**Figure 3 sensors-21-04947-f003:**
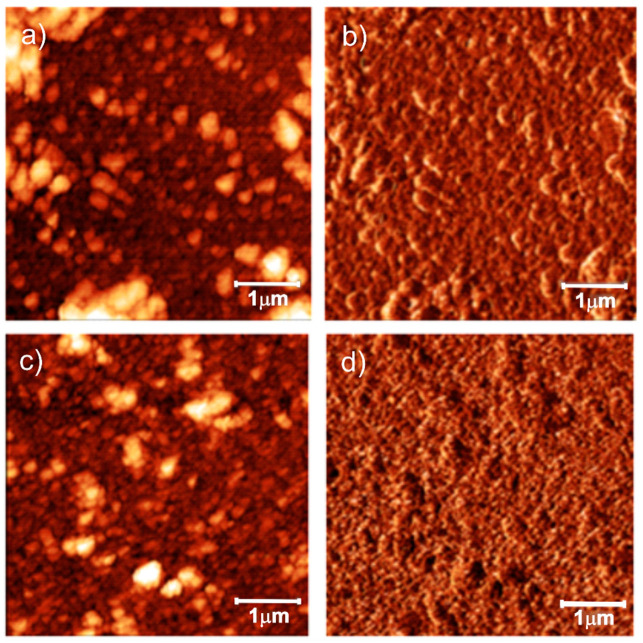
AFM topography (left) and force mode (right) images acquired from (**a**,**b**) PANI and (**c**,**d**) rGO-PANI.

**Figure 4 sensors-21-04947-f004:**
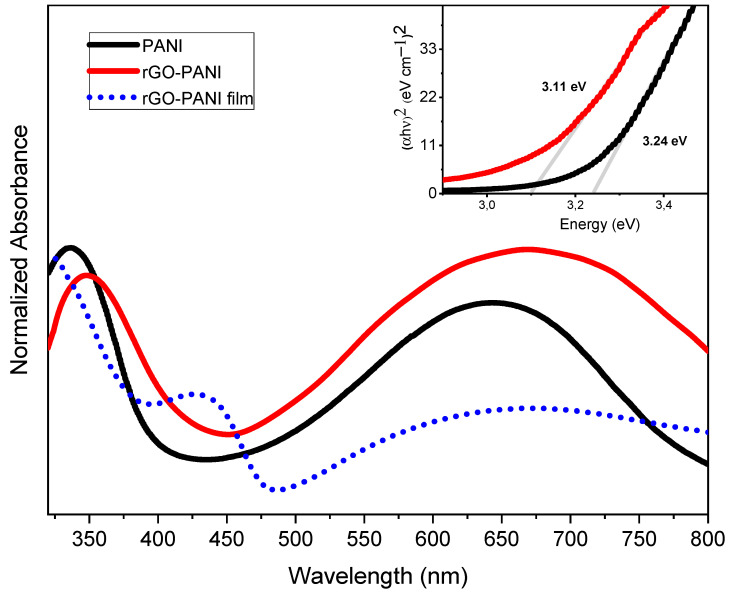
UV-vis spectra acquired from PANI (solution) and rGO-PANI (suspension and film). Inset: Tauc plots for PANI and rGO-PANI (suspensions).

**Figure 5 sensors-21-04947-f005:**
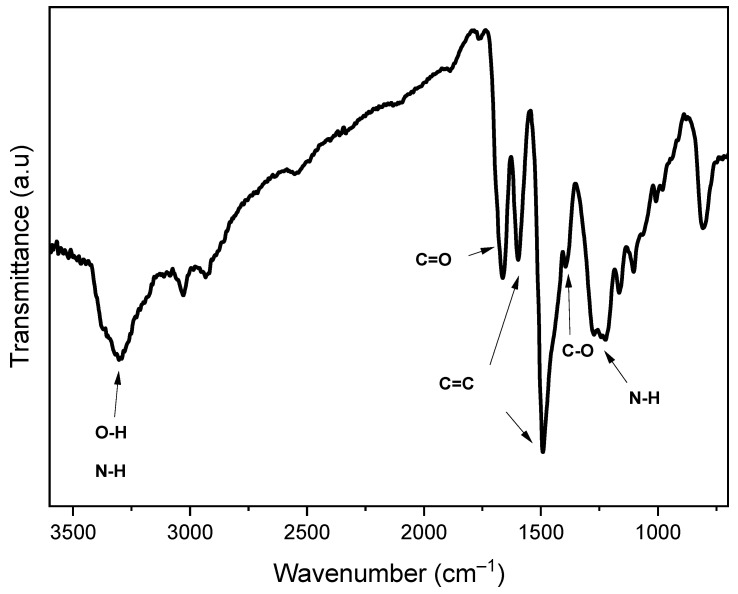
FTIR spectrum of rGO-PANI film.

**Figure 6 sensors-21-04947-f006:**
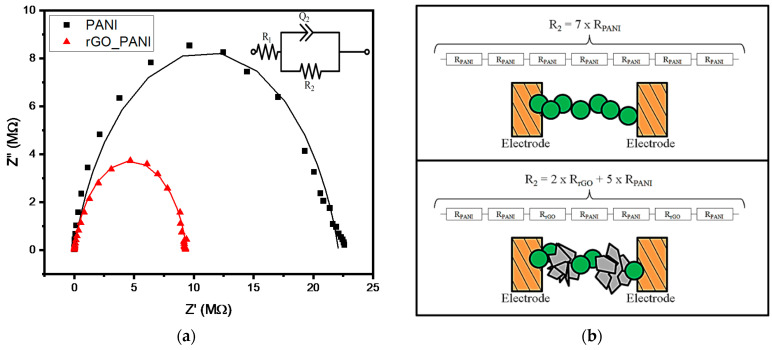
Nyquist plot (**a**) with experimental data (symbols) and fit from the equivalent circuit (dashed lines), inset: equivalent circuit. On (**b**), a schematic representation of PANI and rGO-PANI between the electrodes.

**Figure 7 sensors-21-04947-f007:**
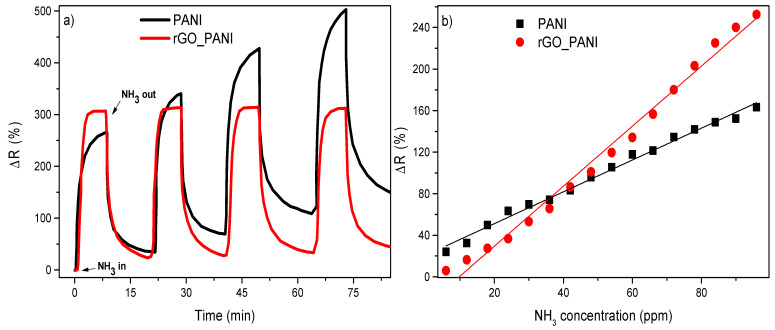
(**a**) Sensitivity versus time (at 600 ppm of NH_3_) and (**b**) calibration curves.

**Table 1 sensors-21-04947-t001:** Main parameters from NH_3_ sensors reported in literature.

Sensing Layer	Sensitivity (at 100 ppm) (%)	Detection Range (ppm)	Response Time (s)	Recovery Time (s)	Reference
rGO-PANI	250	5–600	97	680	This work
rGO-PANI	350	10–100	20	27	[38]
CuTSPc@3D-(N)GF	17	1–1000	150	60	[62]
Graphene_PEDOT:PSS	7	1–1000	180	300	[63]
PANI@SnO_2_	29	10–100	-	-	[15]
PANI@Cu	90	1–100	-	-	[64]
Graphene_PANI	3.5	1–6400	50	23	[18]
PANI	17	5–1000	40	191	[65]

## Data Availability

The data presented in this study are available in article.

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
