# Peer review of "NH3 Sensor Based on rGO-PANI Composite with Improved Sensitivity"

_sensors, 2021, doi:10.3390/s21154947_

Round 1

Reviewer 1 Report

In this paper the author mentioned a reduced graphene oxide and poly(aniline)composite,which resulted in rGO clusterd between PANI chains. This composite was processed as thin film onto intergitated electrode array and used as the sensitive layer for ammonia gas, presenting sensibility of 250% at 100 ppm with response times of ca. 96 s. However, some severe issues should be addressed before proceeding further.

  1. In the first paragraph of the introduction, the author mentioned an ammonia “slip”. Can the author point out where the ammonia “slip” came from.
  2. The layout of the first paragraph on page 5 is different from that of the rest of the article
  3. In Figure 3, the author can adjust the current color appropriately to increase the contras
  4. How about the long-term stability?
  5. It is interesting that the author only plotted the response curve in Fig. 6a and b. Why not present the recovery curve?
  6. Raman scattering for PANI is recommended to test.
  7. BET test was suggested to better illustrate the surface area enhanced by introducing RGO.
  8. The author investigated the nanostructured materials enabled gas sensor. Some related article may enrich the background and concept in the introduction as references: Adv. Funct. Mater., 2021, 31, 2010962, Adv. Funct. Mater. 2018, 28, 1704112ï¼›Joule 2017, 1, 480.

Reviewer 2 Report

The introduction can be made stronger by adding more information regarding using RGO.

The basic sensing mechanisms has to be been explained. 
 Overall the gas sensing results can be explained better. 

Specific comments

Main reason for using RGO should be highlighted and why not Graphene oxide can be used. 

In figure 2 the scale bar can be highlighted. 

The nyquist plot and the circuit analysis results should exlpained more. 

Reviewer 3 Report

Dear Authors,

I think that the manuscript (MS) must be necessarily improved. In the present form, the MS looks like an incomplete report and not a Communication on Sensors - MDPI. It cannot be published as it is, absolutely.

The Authors present a sensor with an high sensitivity toward NH3 for indoor applications. My first and more important question is: why don’t you have measured (and reported here) the influence of humidity on the sensor responses?

My second point is about the fact that it is not clear because it is not well documented what is the novelty of the work.

The Authors must declare in a comprehensive and clear way the names of their samples in the whole MS.

The MS needs a whole revision of the English wording also made with the consciousness of what you are saying.

The MS contains many style errors (lines 80, 93, 110, 119, 121, 144-153, 157, etc.).

The results are not suitably described and discussed.

Title

Title is too generic.

Keywords

Too few and too generic.

Abstract

Why did you write poly(aniline) with aniline inside brackets?

Lines 17-20: Write the sentence in a more clear way.

You must improve the abstract with more information about your work and the peculiarities of your research.

Introduction

Lines 58-61: you must rewrite this sentence in a more comprehensive way and change reported at with reported in.

In the introduction (and in the whole MS) you do not summarize the others rGO-PANI based sensors for ammonia, as well as you don’t make the comparison with those in the (absent) discussion section.

At the end of the introduction, you describe in a not sufficient way your work.

Materials and Methods

Line 78: please clarify (wt:wt) because it is a ratio but you don’t give also the value.

Line 79: please describe the FR-1/metal substrates.

Line 91: sping coating?

Line 110: it is most common the term density.

Results without “.” 

Line 115: are you confused? you prepared PANI pristine films for comparing the characteristics/properties of the PANI pristine films and those of PANI functionalized with rGO films and not to allow the comparison.

Line 119: you must declare how you calculate the average diameter of not nano-particles, 450 nm is not nano.

What is the utility of AFM analysis?

Lines 122-124: please correct.

About the figure:

Figure 1 lacks of tags a, b, c, d.

Figure 3 has a bad quality.

Figure 5 and its description must be improved.

Figure 6b why PANI curve marker was changed ? Are you sure of the content of the figure 6b, and the y axis?

Figure 6c is not described.

Figure 6d are you sure of your description in lines 204-206? I am not.

Why don’t you calculate the response and recovery times?

Discussion and Conclusions

The discussion section is completely absent, and cannot be substituted by Table 1.

The conclusions are very poor and of course insufficient.

Lines 210-211: The sentence is a non-sense.

Line 211: “due” needs to …

Best regards

Round 2

Reviewer 2 Report

The authors answered the comments. 

Author Response

The authors acknowledge the reviewer by the previous comments.

Reviewer 3 Report

Dear Authors,

I think that the manuscript (MS) could be published in present form after some minor revisions.

I put my notes and suggestions (highlighted in blue) in the uploaded file named: sensors-1239070-peer-review-v2_blue.pdf

Best regards

Author Response

The authors acknowledge the reviewer by the comments.

Line 41- The sample name has been edited.

Figure 1 was edited and replaced.

Line 127 – The dot has been removed.

Figure 2 – The letters “a) and b)” were included in Figure 2.

Lines 147-150 – The text has been edited, the word “presented” was replaced by synonymous.

Line 188 - The sample name has been edited.

Page 8, line 220 – The parameters measured from PANI based sensor have been included.

Figure 7 – Legend has been edited.

Line 235 – “NH3” was included.

Table 1 - % symbol was included.

Subsection 4- Discussion was included.